# Extraction Methods, Chemical Characterization, and In Vitro Biological Activities of *Plinia cauliflora* (Mart.) Kausel Peels

**DOI:** 10.3390/ph16081173

**Published:** 2023-08-17

**Authors:** Mariana Moraes Pinc, Mariana Dalmagro, Elton da Cruz Alves Pereira, Guilherme Donadel, Renan Tedeski Thomaz, Camila da Silva, Paula Derksen Macruz, Ezilda Jacomassi, Arquimedes Gasparotto Junior, Jaqueline Hoscheid, Emerson Luiz Botelho Lourenço, Odair Alberton

**Affiliations:** 1Laboratory of Preclinical Research of Natural Products, Paranaense University, Umuarama 87502-210, Paraná, Brazil; mariana.pinc@edu.unipar.br (M.M.P.); mariana.dal@edu.unipar.br (M.D.); elton.221872@edu.unipar.br (E.d.C.A.P.); donadel425@gmail.com (G.D.); renan.thomaz@edu.unipar.br (R.T.T.); ezilda@prof.unipar.br (E.J.); jaquelinehoscheid@prof.unipar.br (J.H.); emerson@prof.unipar.br (E.L.B.L.); 2Department of Technology, State University of Maringá, Umuarama 87506-370, Paraná, Brazil; camiladasilva.eq@gmail.com; 3Department of Chemical Engineering, State University of Maringá, Maringá 87020-900, Paraná, Brazil; pauladmacruz@gmail.com; 4Laboratory of Cardiovascular Pharmacology (LaFaC), Faculty of Health Sciences, Federal University of Grande Dourados, Dourados 79804-970, Mato Grosso do Sul, Brazil; arquimedesjunior@ufgd.edu.br

**Keywords:** infusion, Myrtaceae, vortex extraction, standardization, antioxidant

## Abstract

*Plinia cauliflora* (Mart.) Kausel, popularly known as jabuticaba, possesses bioactive compounds such as flavonoids, tannins, and phenolic acids, known for their antioxidant, antibacterial, wound healing, and cardioprotective effects. Therefore, this study aimed to standardize the *P. cauliflora* fruit peel extraction method, maximize phenolic constituents, and evaluate their antioxidative and antimicrobial effects. Various extraction methods, including vortex extraction with and without precipitation at 25, 40, and 80 °C, and infusion extraction with and without precipitation, were performed using a completely randomized design. Extraction without precipitation (E − P) showed the highest yield (57.9%). However, the precipitated extraction (E + P) method displayed a yield of 45.9%, higher levels of phenolic derivatives, and enhanced antioxidant capacity. Major compounds, such as D-psicose, D-glucose, and citric acid, were identified through gas chromatography–mass spectrometry (GC-MS) analysis. Ultra-high-performance liquid chromatography–tandem mass spectrometry (UHPLC-MS/MS) analysis identified citric acid, hexose, flavonoids, tannins, and quercetin as the major compounds in the extracts. Furthermore, the extracts exhibited inhibitory effects against *Bacillus subtilis*, *Staphylococcus aureus*, *Pseudomonas aeruginosa*, and *Escherichia coli* bacteria. In conclusion, the E + P method efficiently obtained extracts with high content of bioactive compounds showing antioxidant and antimicrobial capacities with potential application as a dietary supplement.

## 1. Introduction

*Plinia cauliflora* (Mart.) Kausel, popularly called jabuticaba, has globose berry fruits up to 3 cm in diameter, with white mucilaginous and bittersweet pulp mainly composed of sugars. These fruits contain one to four seeds and have a dark reddish peel, with promising effects on human and animal health [1,2]. *P. cauliflora* peel extracts have been shown antioxidant [3], antimicrobial [4], and wound-healing properties [5]. Moreover, cardioprotective [6] and hepatoprotective effects [7] were described. Phytochemical analyses predominantly show phenolic compounds, including tannins and organic acids [8].

According to Palozi et al. [3], hydroethanolic extracts of *P. cauliflora* peel contain ellagic acid, gallic acid, O-desoxy-hexosyl quercetin, and O-hexosyl cyanidin anthocyanins. In the same study, preclinical tests confirmed the safety of the pharmacological use of these extracts. Owing to their beneficial health effects and lack of toxicity, *P. cauliflora* peel extracts can be used for daily oral dietary supplementation [8].

The effects of bioactive compounds on health depend on the qualitative and quantitative characteristics and the synergism between these substances, which vary depending on the extraction method used [9]. Therefore, an efficient extraction process can maximize the extraction of bioactive compounds, prevent their degradation, use environmentally friendly technologies, and produce low-cost raw materials [10].

The proper choice of solvent is crucial in producing extracts on an industrial scale, especially when seeking more remarkable purification, requiring inserting a precipitating agent for polysaccharides. In this regard, water is an effective solvent for extraction, allowing easy binding of polysaccharides to water molecules through hydrogen bonds, ensuring their solubility and successful extraction. On the other hand, ethanol is commonly used as a precipitating agent because it reduces the polarity of water, rendering polysaccharides insoluble [11]. Additionally, because of its renewable origin (from sugarcane) and its classification as generally recognized as safe (GRAS), ethanol is suitable for green chemical extraction [12].

Thus, this study aimed to determine the effects of different methodologies on the extraction performance of phenolic derivatives of *P. cauliflora* fruit peel extracts and identify the phytochemical components and their antioxidant and antimicrobial properties. Overall, this study may pave the way for bioprospecting formulations enriched with *P. cauliflora* for dietary supplementation.

## 2. Results

### 2.1. Yield of the Extracts

The method with the highest yield (Figure 1) was extraction without precipitation (E − P) (57.9%), followed by vortex extraction without precipitation at 40 °C (T − P40) (52.8%) and extraction with precipitation (E + P) (45.9%). The extraction method with the lowest yield was vortex extraction with precipitation at 80 °C (T + P80) (28.9%).

### 2.2. Characterization by GC-MS and UHPLC-MS/MS

Gas chromatography coupled with mass spectrometry (GC-MS) identified 83 compounds in the *P. cauliflora* extracts (Table 1) and classified them into the following categories: phenols and derivatives; sugars and derivatives; carboxylic acids and derivatives; alkaloids, fatty acids, glycerolipids, hydroxy acids, and derivatives; keto acids and derivatives; organooxygen compounds; and quinones. The major compounds identified were D-psicose 5TMS, D-glucose 5TMS, citric acid 3TMS, and glycerol 3TMS. GC–MS analysis revealed a predominance (%) of sugars and derivatives, with D-psicose 5TMS as the main constituent in T + P25, T + P40, and E + P, D-(−)-fructofuranose; pentakis (trimethylsilyl) ether (isomer 1) as the main constituent in T-P80; and D-glucose 5 TMS as the main compound in T + P25, T + P40, and E − P. The compounds were identified by searching the library database of Spectra NIST Mass Spectral Library (version 2014), and the main fragments of major compounds were compared with the literature, as shown in Table A1.

A total of 27 compounds were identified using ultra-high-performance liquid chromatography–tandem mass spectrometry (UHPLC-MS/MS) in the positive and negative modes (Table 2 and Table 3). These compounds are classified as flavonoids, phenolic acids, tannins, and their derivatives; sugars and their derivatives; carboxylic acids and their derivatives; and alkaloids. UHPLC-MS/MS analysis in negative mode revealed the predominance of carboxylic acids and their derivatives, with citric acid being the major compound. In contrast, the positive mode showed a predominance of flavonoids, with quercetin being the major compound, followed by sugars and derivatives, with hexose and di-hexoside being the most abundant. The ion chromatogram and MS and MS/MS spectra were visualized using Data Analysis 4.3 software and the compounds were identified based on literature data, according to Romão et al. [6].

### 2.3. Quantification of Total Phenolic Compounds and Flavonoids

Regarding the quantification of phenolic constituents (Table 4), it was observed that E + P extraction significantly increased the total phenolic content (TPC) and total flavonoid content (TFC) (115.59 ± 1.79 µg GAE gext^−1^ and 6.95 ± 0.04 µg QE gext^−1^, respectively).

The E + P method significantly increased the scavenging capacity of 2,2-Diphenyl-1-picrylhydrazyl (DPPH) radical and 2,2-Azinobis (3-ethylbenzthiazoline-6-sulfonic acid) (ABTS) radicals (Table 5). However, there was no significant difference between the E − P and E + P extracts in terms of the reducing power of the ferric (III)/tripyridyltriazine (FRAP) complex. Similarly, T − P40 exhibited the lowest antioxidant activity.

### 2.4. Evaluation of Antimicrobial Activity

All extracts showed the ability to inhibit both Gram-positive and Gram-negative bacteria, but did not exhibit antifungal activity against *Candida albicans* at the evaluated concentrations (Table 6). Overall, E + P exhibited the lowest minimum inhibitory concentration (MIC), followed by T + P25.

Hierarchical clustering was employed to illustrate the variability in the quantification of phenolic compounds and flavonoids and the antioxidant capacity of the methodologies used for the production of *P. cauliflora* extracts (Figure 2). The samples were grouped into four main clusters and displayed in a dendrogram obtained using the Unweighted Pair Group Method with Arithmetic Mean (UPGMA) using unweighted arithmetic averages.

The first cluster included the E + P method, indicating that the results obtained were more significant in terms of the quantity of phenolic compounds and flavonoids, as well as the antioxidant capacity of this sample, compared with the other methods.

Principal component analysis (PCA) allowed for the joint evaluation of all variables. Multivariate analysis was applied to assess the antioxidant capacity of the extracts and quantify TPC and TFC, considering all preparation methods. PCA showed a total variance of principal components of 90.52%, of which 65.46% was explained by PC1 and 25.06% by PC2 (Figure 3).

The E + P and T − P40 methods were relatively displaced compared with the other methods, indicating more and less significant responses, respectively.

## 3. Discussion

Fruits are the main dietary source of polyphenols, which, owing to intrinsic and extrinsic factors, exhibit varied compositions of these constituents in terms of quantity and quality [13]. High temperatures increase the diffusion rate and solubility of compounds. However, depending on the conditions employed, degradation or partial removal of the active compounds may occur. Additionally, different levels of complexity in the structure of phenolic compounds lead to variations in their sensitivity to extraction conditions. Thus, the yield and composition of the extracts, and consequently their properties, depend on the extraction conditions [14].

The highest yield was obtained using the E − P extraction method. This can be attributed to the non-removal of gums, mucilages, and proteins that occur in other methods in the presence of precipitates through ethanol addition. Consequently, the extraction process was shortened without the need for a second filtration step. However, higher yields in the extraction process do not necessarily indicate higher efficiency [15]. Furthermore, difficulties in handling lyophilized extracts are encountered in these methods because, despite generally having a higher mass without precipitation, the extracts exhibit viscous and adhesive properties. In a previous study [16], the plant-to-solvent ratio was 1:2, whereas, in our study, this ratio was adjusted to 1:10. The obtained results suggest that this modification in the plant-to-solvent ratio also affects the extraction yield, supporting the assertions made by the same authors.

In general, E + P proved to be the most effective compared to others because the extraction temperature employed increased the diffusion rate and enhanced the solubilization of phenolic compounds and flavonoids, concurrently with the precipitation of proteins and polysaccharides by the addition of ethanol. This explains the lower yield of this extraction method compared to that of E − P, although it exhibited superior antioxidant capacity.

The assessment of antioxidant capacity is frequently performed using methods such as FRAP in conjunction with other techniques. Antioxidants derived from natural sources such as fruit peels play a vital role in free radical scavenging and inhibiting iron and copper chain reactions [17]. In this study, the results were directly proportional to the antioxidant capacity exhibited by each extraction process employed to prepare extracts from mature *P. cauliflora* fruit peels. Furthermore, a remarkable correlation was observed between total phenolic compounds and flavonoids.

Different phenolic compounds scavenge different types of free radicals. Flavonoids, tannins, and condensed tannins contribute to the antioxidant capacity of ABTS, whereas anthocyanins contribute to that of DPPH [18]. Baldin et al. [19] found a lower antioxidant effect in FRAP and DPPH assays for microencapsulated aqueous jabuticaba extracts than in the present study. Different DPPH values occur because of the extraction method, solvents used, and drying process, which can concentrate the components of the extract, and consequently explain the different levels of phenolic compounds observed for each extraction method used in each study [19]. Lenquiste et al. [20] analyzed the aqueous and methanolic extracts of lyophilized jabuticaba peels and found lower antioxidant capacities for ABTS, DPPH, and FRAP than that of E + P.

The use of different chromatographic techniques provides a comprehensive metabolic profile that aids in the identification and quantification of the major chemical markers of plant species. Generally, GC–MS is better for identifying primary metabolites, such as amino acids, fatty acids, carbohydrates, and organic acids [21]. However, UHPLC-MS/MS is more suitable for identifying secondary metabolites, such as alkaloids, saponins, phenolic acids, terpenoids, flavonoids, and glycosides [22].

Through GC-MS and UHPLC-MS/MS analyses, it was possible to obtain a comprehensive view of the chemical composition of the extracts, highlighting the presence of bioactive compounds such as flavonoids, sugars, tannins, and organic acids, which were responsible for the observed antimicrobial and antioxidant activities in this study. However, it is important to acknowledge that these compounds’ measures were performed semi-quantitatively, which may have limitations in providing absolute concentrations.

Some studies have suggested that D-psicose, a major compound found by GC-MS, can suppress hyperglycemia by exhibiting hypolipidemic and antioxidant activities [23]. Furthermore, it has been indicated as a potential protective agent against type 2 diabetes [24] and its complications, such as cardiovascular diseases and hepatic steatosis, making it an ideal substitute for sucrose [23]. D-psicose is a monosaccharide that can be enzymatically produced from D-glucose via D-fructose catalyzed by D-xylose isomerase and D-tagatose 3-epimerase. However, it is a rare sugar found in nature [25].

Evidenced in large quantities in this study, citric acid plays a vital role in determining the degree of ripeness of the fruit, influencing its flavor and the balance between alkalinity and acidity. However, the activities of these acids as bioactive compounds are variable and not fully understood [26].

According to a study by Gomez-Delgado et al. [27], citric acid has significant benefits in regenerative processes due to its ability to promote the increased release of transforming growth factor beta (TGF-β1). Citric acid is widely used as a flavoring agent and preservative in the food and beverage industry. However, the mechanism of the antimicrobial action of citric acid is not wholly understood [28]. Furthermore, In, Kim, Kim, and Oh [29] have reported that citric acid exhibits weak antibacterial activity against foodborne pathogens.

Quercetin, a flavonoid found in fruits and vegetables, and evidenced as one of the major compounds present in jabuticaba, has various beneficial effects on human health [30]. Scientific studies have investigated the effects of quercetin supplementation on antioxidant and anti-inflammatory activity [31,32]. Furthermore, its potential for cancer prevention [33] and its inhibitory effect against different strains of methicillin-resistant *Staphylococcus aureus* (MRSA) were also investigated [34].

Regarding antimicrobial activity, the microdilution technique has been used in a limited number of studies with *P. cauliflora* extracts, as is the case of the present study. Oliveira et al. [14] evaluated jabuticaba peel extracts using four solvents: acetone, water, ethanol, and methanol. None of the tested extracts showed efficacy in inhibiting the growth of the Gram-negative bacteria *E. coli* and *Salmonella choleraesuis*. However, the extracts exhibited minimum inhibitory concentrations (MICs) of 250 µg/mL against *P. aeruginosa* [14].

From another perspective, Fleck et al. [35] demonstrated the antimicrobial activity of the aqueous extract of jabuticaba peel against *S. aureus*, *B. cereus*, and *E. coli* with minimum inhibitory concentrations of 11.22 ± 0, 8.42 ± 2.52, and 2.80 ± 0.11 mg mL^−1^, respectively. According to Fleck [34], this activity may be related to the anthocyanins and phenolic acids in fruit peels.

The E + P method exhibited higher antimicrobial activity consistent with the results described in Table 6, although some metabolites were not significantly detected. This observation can be attributed to the precipitation, concentration, and synergism of the bioactive compounds. One notable example is the ability of phenolic compounds to affect the functioning of bacterial cells in various ways. In addition to interfering with enzymatic activity, they can also influence bacterial metabolic processes by forming complexes with metal ions [36].

Anthocyanins are another group of bioactive compounds that stand out for their antimicrobial action mechanisms. Studies conducted by Cisowska and Hendrich [37] demonstrated that the inhibitory effect of anthocyanins present in fruit peels arises from multiple mechanisms and synergies with other compounds, such as weak organic acids, phenolic acids, and their different chemical forms, resulting in various membrane and intercellular interactions that contribute to the antimicrobial action observed in this study.

Therefore, it is important to highlight the use of jabuticaba peel as a source of bioactive compounds that can contribute to the reduction in synthetic chemical use and benefit consumer health. Supplementation with phytochemical complexes based on their proportion and synergism results in more affordable and accessible products for healthy individuals who do not require a medical prescription. Additionally, the use of peel reduces resource wastage and contributes to environmental sustainability by preventing the accumulation of organic residues [14,38,39].

## 4. Material and Methods

### 4.1. Plant Material

Ripe fruits of *P. cauliflora* were collected from a rural property in the city of Esperança Nova (Paraná, Brazil, 23.719864, −53.802104). The specimen was deposited in the herbarium of Universidade Paranaense under number 339 (SisGen number: A672209). After collection, the fruits were washed with running water, and their peels were manually removed and dried by forced air circulation for 5 d at 45 °C. Subsequently, they were pulverized in a knife mill and stored in plastic bags at 2–8 °C until further use.

### 4.2. Extraction Processes

A plant material-to-solvent ratio of 1:10 (*w*/*v*) was used to produce the extracts. The effects of temperature, protein precipitation, and carbohydrate precipitation in 95% ethanol and vortex extraction were evaluated, resulting in eight extracts. The study design is illustrated in Figure 4.

Powdered dried peels (25 g) were initially submitted to contact with filtered water (250 mL) at different temperatures (25, 40, and 80 °C) for 1.5 h, followed by vortexing for 5 min at a rate of >2000 rpm according to Islam et al. [16] with modifications. After filtering the solid residues, the first aliquot was frozen and lyophilized. The second aliquot was treated at an extract-to-ethanol ratio of 1:3 (*v*/*v*). The ethanol (95% ethyl alcohol) addition reduced the polarity of water, causing the in-solubilization of polysaccharides and proteins. Subsequently, the precipitate was removed by filtration, the solvent was removed using a rotary evaporator (Nova Ética, São Paulo, Brazil), and the resulting fraction was lyophilized (JJ Científica, model LJJ02, São Paulo, Brazil).

The plant material was placed in water at 90 °C for 6 h for preparing infusion. Subsequently, the sample was filtered with or without precipitation, as previously described. The extraction yield was calculated as the ratio of the extracted mass to the initial mass of the dried peels.

### 4.3. Phytochemical Evaluation

#### 4.3.1. GC-MS Analysis

For chemical identification by GC-MS before analysis, samples (~40 mg) were derivatized using bis(trimethylsilyl) trifluoroacetamide with trimethylchlorosilane (BSTFA/TMCS) (Sigma-Aldrich, Saint Louis, MO, USA) and pyridine (Anidrol, Diadema, Brazil) (200:200 µL) at 90 °C for 1 h in an oven with air circulation, as indicated by Canini et al. [40], and then diluted in ethyl acetate (Anidrol, Diadema, Brazil) to a final volume of 1 mL. The solution was analyzed by gas chromatography using a mass spectrometer (Shimadzu, GCMS-QP2010 SE, Tokyo, Japan) equipped with an automatic injector (AOC-20i) and capillary column SH-Rtx-5MS (Shimadzu, 30 m × 0.25 mm × 0.25 µm, Tokyo, Japan). Helium (White Martins, purity > 99%) was used as the carrier gas at a flow rate of 1.0 mL min^−1^ with a split ratio of 1:30, and the injection volume was 3 μL. The column temperature was initially programmed to be 100 °C, heated at 4 °C min^−1^ to 280 °C, and then heated at 10 °C min^−1^ to reach a final temperature of 300 °C. The temperature of the injector and MS interface was maintained at 250 °C. Mass spectra were recorded at 70 eV with a mass range of *m*/*z* 35–550. The compounds were identified by searching the library database of Spectra NIST Mass Spectral Library (version 2014). The relative abundance of each metabolite was calculated by multiplying the individual area of the compound by 100 and then dividing it by the total area of all identified compounds in the sample.

#### 4.3.2. UHPLC-MS/MS Analysis

Jabuticaba extracts were analyzed using a UHPLC-MS/MS system (Shimadzu Nexera X2, Japan) coupled to a Q-TOF Impact II mass spectrometer (Bruker, Germany). The extracts were prepared at a concentration of 1 mg mL^−1^ in methanol–water (1:1, *v*/*v*), filtered using polytetrafluoroethylene (PTFE) filters (Millex, 0.22 mm × 13 mm, Millipore), and injected at a volume of 2 µL.

The UHPLC-MS/MS system was equipped with a UPLC CSH C18 column (Waters, USA, 1.7 μm, 2.1 × 100 mm). The mobile phase was a mixture of solvent A (water with 0.1% formic acid, *v*/*v*) and solvent B (acetonitrile with 0.1% formic acid, *v*/*v*) at a flow rate of 0.250 mL min^−1^. The A:B gradient used was as follows: 3% B from 0 to 1 min, 50% B from 1 to 10 min, 95% B from 10 to 15 min, 95% B from 15 to 19 min, 3% B from 19 to 21 min, and maintained at 3% B from 21 to 25 min at 40 °C, with the last four minutes dedicated to column reconstitution for the subsequent analysis. For negative analysis, solvent mixtures A (water with 0.1% formic acid, *v*/*v*) and B (acetonitrile) were used. The gradient mixture for negative analysis was the same as that for positive analysis.

A Q-TOF Impact II mass spectrometer with an electrospray ionization source was used in the auto MS/MS acquisition mode. The acquisition rate was 5 Hz (MS and MS/MS), and the tuning range was *m*/*z* 120–1200. Mass spectra were collected using electrospray ionization (ESI) in positive and negative ion modes, with a capillary voltage set at 3.50 kV, source temperature of 200 °C, and a desolvation gas flow rate of 9 L min^−1^.

The ion chromatogram and MS and MS/MS spectra were visualized using Data Analysis 4.3 software and compared with the existing literature. This method was developed based on that described by Tolouei et al. [41].

### 4.4. Spectrophotometric Analysis

The extracts were prepared at a concentration of 1000 µg mL^−1^ and evaluated in independent triplicates. Total phenolic content (TPC) was quantified using the Folin–Ciocalteu method [42] on a UV/Vis spectrophotometer (Kasuaki Model IL-592) at 765 nm. The results were calculated based on the calibration curve for gallic acid (Sigma-Aldrich) (straight-line equation: y = 14.269x + 65.544; coefficient of correlation R^2^ = 0.9905). The results are expressed as µg equivalents of gallic acid per gram of extract (µg EAG gext^−1^).

The total flavonoid content (TFC) was determined based on the method described by Woisky and Salatino (1998) [43], and absorbance was measured at 425 nm using a spectrophotometer (Kasuaki Model IL-592). A standard calibration curve of quercetin (y = 81.561x − 126.41; R^2^ = 0.9966) was plotted for quantification, and the results were expressed as µg equivalents of quercetin per gram of extract (µg QUE gext^−1^).

### 4.5. Antioxidant Analysis

The extracts were prepared at a concentration of 1000 µg mL^−1^ and evaluated for their scavenging capacity against DPPH and ABTS radicals and their reducing capacity using the FRAP assay, in independent triplicates.

The DPPH assay was performed as described by Silveira et al. [44]. A calibration curve (y = −0.5771x + 673.63; R^2^ = 0.9968) was plotted using Trolox (Sigma-Aldrich), and the DPPH radical scavenging capacity was expressed in μM Trolox equivalents (μM Trolox).

ABTS assays were conducted as described by Re et al. [45]. A calibration curve was plotted using Trolox (y = −0.2465x + 750.59; R^2^ = 0.9914), and the ABTS free radical scavenging capacity was expressed in μmol of Trolox per gram of extract (μmol Trolox gext^−1^).

The FRAP analysis was performed according to the methodology described by Santos et al. [46]. To determine the antioxidant capacity, a calibration curve for ferrous sulfate (Sigma-Aldrich) was plotted (y = 0.6188x − 96.833; R^2^ = 0.9926), and the results were presented as µmol of Fe^2+^ per gram of extract (µmol Fe^2+^ gext^−1^).

### 4.6. In Vitro Antimicrobial Activity

The minimum inhibitory concentration (MIC) was determined in triplicate following a previously described methodology [47]. The test was conducted against the microorganisms *Bacillus subtilis* (CCCD B005), *Escherichia coli* (ATCC 25922), *Staphylococcus aureus* (ATCC 12026), *Pseudomonas aeruginosa* (ATCC 9027), and *Candida albicans* (ATCC 10231).

The extracts were added to microplates containing Brain Heart Infusion (BHI) broth at concentrations ranging from 125 mg mL^−1^ to 1.95 mg mL^−1^. Microbial suspensions were prepared in sterile water at a McFarland scale of 0.5. The microplates were incubated at 36 °C for 24 h for bacteria and 27 °C for 48 h for yeast. Subsequently, a 2% solution of 2,3,5-triphenyltetrazolium chloride (TTC) was added and incubated for 2 h. A reddish color indicates microbial growth.

### 4.7. Statistical Analysis

The results of different extraction methods were subjected to Levene’s test for data homogeneity, followed by an analysis of variance (ANOVA). Means were compared by Duncan’s test (*p* ≤ 0.05) using SPSS software (version 22.0; SPSS Inc., Chicago, IL, USA).

Cluster analysis (CA) and principal component analysis (PCA) were performed to discriminate the composition of the extracts based on the different preparation methodologies, and the variables were analyzed using Statistica v. 13.3.

## 5. Conclusions

The compounds identified in extracts from *Plinia cauliflora* fruit peels, including flavonoids, tannins, and organic acids, reinforce the potential bioactive effects of jabuticaba, including its antioxidant and antimicrobial capacities. Among the extraction methods, extraction with precipitation (E + P) demonstrated higher efficiency when compared to the other extraction methods. It is important to note that this study focused on utilizing phytocomplexes to leverage the synergistic effects among bioactive compounds rather than investigating isolated compounds. This approach is relevant as it may reflect the real-life scenarios of bioactive interactions and potential health benefits when using the whole extract. Therefore, the E + P method emerges as a promising extraction technique for large-scale production in the formulation of food supplements.

## Figures and Tables

**Figure 1 pharmaceuticals-16-01173-f001:**
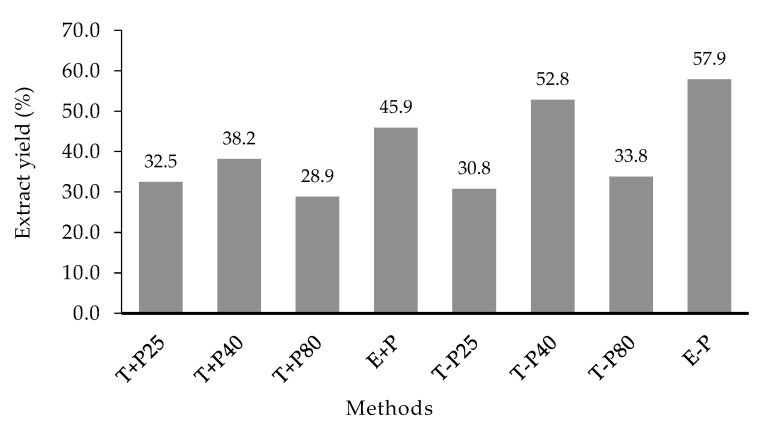
*Plinia cauliflora* extract yields using different extraction methods. T + P25: Vortex extraction with precipitation at 25 °C. T + P40: Vortex extraction with precipitation at 40 °C. T + P80: Vortex extraction with precipitation at 80 °C. E + P: Extraction with precipitation. T-P25: Vortex extraction without precipitation at 25 °C. T − P40: Vortex extraction without precipitation at 40 °C. T − P80: Vortex extraction without precipitation at 80 °C. E − P: Extraction without precipitation.

**Figure 2 pharmaceuticals-16-01173-f002:**
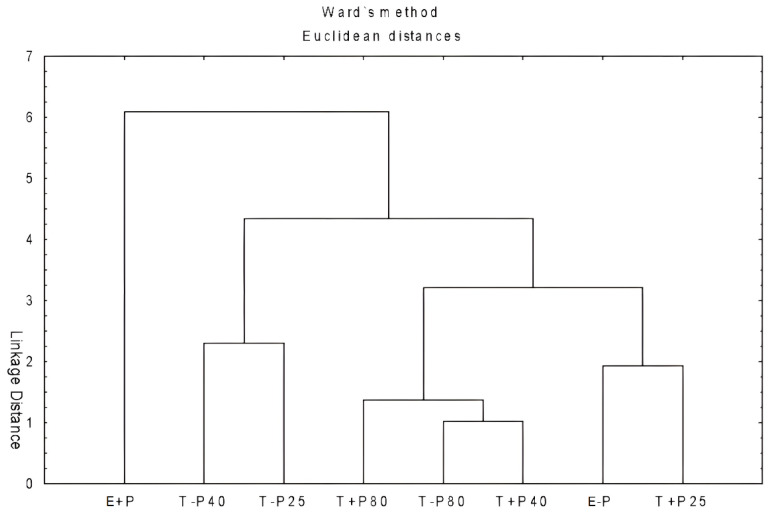
Dendrogram of hierarchical clustering of extracts obtained from *Plinia cauliflora* using different extraction methods, based on data from Table 4 and Table 5. T + P25: Vortex extraction with precipitation at 25 °C. T + P40: Vortex extraction with precipitation at 40 °C. T + P80: Vortex extraction with precipitation at 80 °C. E + P: Extraction with precipitation. T − P25: Vortex extraction without precipitation at 25 °C. T − P40: Vortex extraction without precipitation at 40 °C. T − P80: Vortex extraction without precipitation at 80 °C. E − P: Extraction without precipitation.

**Figure 3 pharmaceuticals-16-01173-f003:**
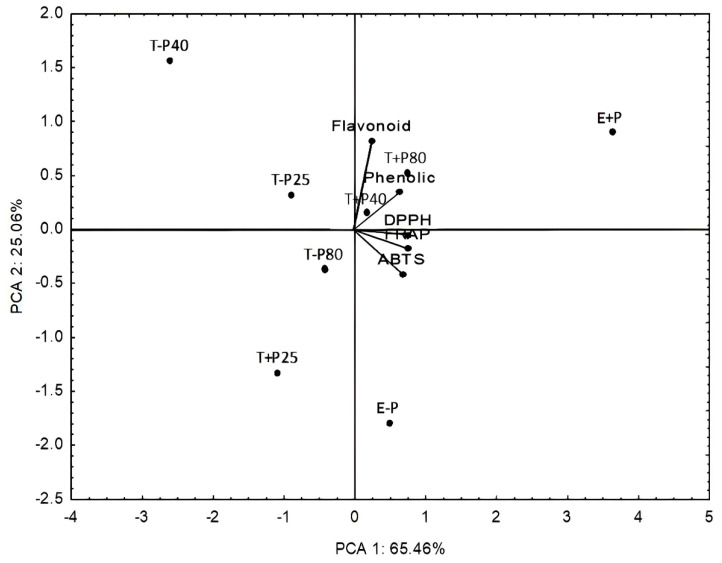
Biplot representation of principal component analysis (PCA) performed on extracts of *Plinia cauliflora* obtained by different extraction methods. T + P25: Vortex extraction with precipitation at 25 °C. T+P40: Vortex extraction with precipitation at 40 °C. T + P80: Vortex extraction with precipitation at 80 °C. E + P: Extraction with precipitation. T − P25: Vortex extraction without precipitation at 25 °C. T − P40: Vortex extraction without precipitation at 40 °C. T − P80: Vortex extraction without precipitation at 80 °C. E − P: Extraction without precipitation.

**Figure 4 pharmaceuticals-16-01173-f004:**
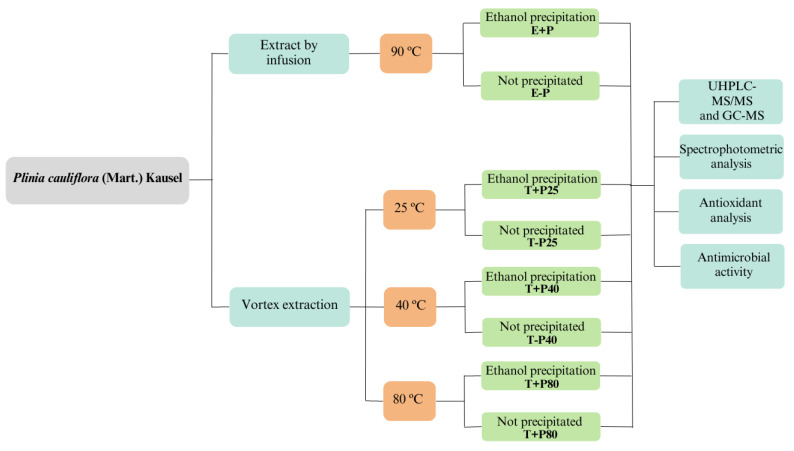
Experimental design for *Plinia cauliflora* extraction.

**Table 1 pharmaceuticals-16-01173-t001:** Phytochemical characterization of *Plinia cauliflora* extracts using gas chromatography coupled with mass spectrometry (GC-MS).

Compound			Relative Composition (%) *							
	**m*/*z**	RT	T + P25	T + P40	T + P80	E + P	T − P25	T − P40	T − P80	E − P
Phenols and derivates			0.47	0.39	0.49	0.66	0.46	0.15	0.82	0.88
Gallic acid, 4TMS	458	27.28	0.47	0.39	0.49	0.66	0.46	n.d **	0.82	0.88
4-Hydroxybutanoic acid, 2TMS	248	7.49	n.d	n.d	n.d	n.d	n.d	0.15	n.d	n.d
Sugars and derivates			70.86	69.70	55.84	77.50	60.39	57.92	67.09	68.27
α-D-Glucopyranose, 5TMS	540	26.01	n.d	n.d	n.d	n.d	n.d	9.77	n.d	n.d
β-D-Galactofuranose, 1,2,3,5,6-pentakis-O-(TMS)	540	24.84	n.d	n.d	2.30	n.d	n.d	n.d	1.74	5.78
3,8-dioxa-2,9-disiladecan-5-one, 2,2,6,6,9,9-hexamethyl	276	22.46	n.d	n.d	n.d	n.d	n.d	0.13	n.d	n.d
D-(−)-Fructofuranose, pentakis(TMS) ether (isomer 1)	540	23.90	n.d	2.80	3.73	15.45	n.d	n.d	15.13	0.75
D-(−)-Fructofuranose, pentakis(TMS) ether (isomer 2)	540	23.92	5.55	n.d	n.d	0.36	12.65	10.52	0.09	3.74
D-(−)-Tagatofuranose, pentakis(TMS) ether (isomer 1)	540	18.56	1.74	0.11	n.d	n.d	n.d	n.d	0.12	3.91
D-(+)-Ribono-1,4-lactone, 3TMS	364	19.51	0.14	0.14	n.d	n.d	0.10	0.21	n.d	n.d
D-Glucose, 5TMS	540	28.31	21.62	19.49	8.76	n.d	11.65	11.07	8.70	16.98
D-Psicofuranose, pentakis (TMS) ether (isomer 1)	540	24.10	n.d	n.d	12.81	1.80	8.80	n.d	15.00	n.d
D-Psicofuranose, pentakis (TMS) ether (isomer 2)	540	23.19	0.10	n.d	n.d	n.d	2.09	0.25	0.12	2.62
D-Xylopyranose, 4TMS	438	26.30	7.91	5.80	0.43	8.60	0.66	2.60	n.d	7.01
Glyceryl-glycoside, TMS	686	35.11	0.08	0.07	n.d	n.d	n.d	n.d	n.d	n.d
Ethyl α-D-glucopyranoside, 4TMS	496	27.09	n.d	n.d	n.d	0.20	0.12	n.d	n.d	n.d
Glucopyranose, 5TMS	540	40.44	n.d	n.d	n.d	8.02	n.d	n.d	n.d	n.d
Deoxyglucose, 4TMS	452	44.94	n.d	n.d	n.d	0.39	n.d	n.d	n.d	n.d
Glucuronolactone, TMS	392	36.52	0.08	n.d	0.18	n.d	n.d	n.d	n.d	n.d
Arabinofuranose, TMS	438	22.94	n.d	n.d	n.d	2.62	n.d	n.d	n.d	n.d
D-Arabinose, 4TMS	438	19.57	n.d	0.09	n.d	n.d	n.d	n.d	n.d	n.d
2-Deoxyribose, 3TMS	350	21.02	n.d	n.d	n.d	n.d	n.d	0.15	n.d	0.08
3-α-Mannobiose	918	43.41	0.41	0.60	0.19	0.26	0.11	0.21	0.25	n.d
Arabinitol, 5TMS	512	37.70	0.07	0.18	n.d	1.82	n.d	n.d	n.d	n.d
Arabinonic acid, TMS	364	22.20	n.d	n.d	2.27	n.d	n.d	2.80	n.d	n.d
D-(−)-Lyxofuranose, TMS	438	42.79	n.d	n.d	n.d	n.d	n.d	n.d	0.19	1.14
D-(−)-Ribofuranose, TMS	438	40.54	0.20	0.14	n.d	n.d	n.d	n.d	n.d	n.d
D-(−)-Tagatose, 5TMS	540	25.77	n.d	n.d	10.44	n.d	n.d	2.34	n.d	n.d
D-(+)-Galactose, TMS	583	27.82	2.48	1.21	n.d	n.d	n.d	n.d	n.d	n.d
D-(+)-Talofuranose, TMS	540	24.84	n.d	5.93	0.24	14.91	2.55	4.76	n.d	n.d
D-(+)-Trehalose, TMS	918	43.60	n.d	n.d	1.01	0.66	1.28	1.16	1.84	n.d
D-(+)-Turanose, TMS	918	42.28	0.16	0.36	n.d	0.49	n.d	n.d	n.d	n.d
D-Gluconic acid, 6TMS	628	28.59	n.d	3.23	n.d	1.72	0.50	n.d	2.56	n.d
D-Mannitol, 6TMS	614	26.74	n.d	n.d	1.25	0.15	n.d	n.d	n.d	n.d
D-Psicose, 5TMS	540	26.01	22.49	22.59	n.d	17.17	10.02	0.17	10.67	16.10
D-Trehalose, TMS	918	41.91	n.d	n.d	n.d	0.23	n.d	n.d	n.d	0.14
D-Xylose, 4TMS	438	27.43	n.d	n.d	0.53	n.d	1.01	1.34	n.d	n.d
Erythritol, 4TMS	410	15.56	n.d	n.d	0.31	n.d	0.27	0.30	0.35	0.06
Gluonic acid, 4TMS	466	26.13	3.25	2.79	3.15	n.d	3.20	4.76	2.28	n.d
L-(−)-Sorbofuranose, TMS	540	42.79	0.14	n.d	n.d	n.d	n.d	n.d	n.d	n.d
L-(−)-Sorbose, 5TMS	540	27.70	n.d	0.19	0.03	2.11	n.d	0.17	n.d	n.d
L-Sorbopyranose, 5TMS	540	25.00	0.73	n.d	n.d	n.d	n.d	n.d	n.d	n.d
Lactulose, TMS	918	42.89	0.50	0.19	n.d	0.34	n.d	n.d	0.16	0.12
Melibiose, TMS	918	44.98	0.48	0.14	n.d	n.d	n.d	n.d	n.d	n.d
Ribonic acid, TMS	526	37.80	0.08	n.d	n.d	n.d	n.d	n.d	n.d	n.d
D-Sorbitol, 6TMS	614	27.00	n.d	n.d	1.41	n.d	0.16	n.d	1.58	n.d
Sucrose, 8TMS	918	41.69	n.d	n.d	n.d	0.21	n.d	n.d	n.d	n.d
Xylitol, 5TMS	512	21.56	2.68	3.66	6.81	n.d	2.99	4.38	6.33	2.73
Xylose, 4TMS	438	25.72	n.d	n.d	n.d	n.d	1.70	n.d	n.d	5.40
β-D-talopyranose, 5TMS	540	26.60	n.d	n.d	n.d	n.d	0.56	0.84	n.d	1.74
Carboxylic acids and derivates			16.29	8.89	24.23	3.51	19.45	14.41	9.50	9.81
Cyclohexanone-3-carboxylic acid	214	14.77	n.d	n.d	n.d	n.d	n.d	0.32	n.d	n.d
3-Butenoic acid, 3-(trimethylsiloxy)-,TMS ester	246	17.30	n.d	n.d	n.d	n.d	n.d	n.d	n.d	0.67
3,4,5-Trihydroxypentanoic acid, tetrakis(TMS)	286	19.89	0.34	n.d	n.d	n.d	n.d	n.d	n.d	n.d
Valeric acid, TMS	364	18.98	0.18	n.d	n.d	0.32	0.12	n.d	n.d	0.28
Acrylic acid, TMS	320	26.48	0.97	1.15	0.22	1.06	0.25	n.d	n.d	1.09
Citric acid, 3TMS	408	23.05	8.70	6.98	23.14	1.73	17.51	8.32	8.61	7.40
Glutaric acid, TMS	364	17.32	n.d	n.d	n.d	n.d	0.49	1.63	n.d	n.d
Methylmalonic acid, 2TMS	540	24.03	5.56	n.d	n.d	n.d	n.d	n.d	n.d	n.d
succinic acid, 2TMS	262	9.60	0.54	0.76	0.87	0.41	1.10	4.14	0.89	0.37
Alkaloids			2.80	2.91	0.97	4.36	0.87	0.91	2.99	0.64
Quininic acid, 5TMS	552	25.15	2.80	2.91	0.97	4.36	0.87	0.91	2.99	0.64
Dioxanes			n.d	1.95	n.d	n.d	2.27	n.d	0.11	11.36
1,3-Dihydroxyacetone dimer, 4TMS	468	22.18	n.d	1.95	n.d	n.d	2.27	n.d	0.11	11.36
Fatty acids			0.09	0.34	0.55	0.08	0.65	1.24	0.67	0.14
Acetin, bis-1,3- TMS ether	278	6.16	n.d	n.d	0.21	n.d	0.32	1.07	n.d	0.12
Linoleic acid, TMS	352	32.51	0.09	0.13	0.13	n.d	n.d	n.d	n.d	n.d
Linoelaidic acid, TMS	352	32.50	n.d	n.d	n.d	n.d	0.15	0.17	0.31	n.d
Palmitic Acid, TMS	328	28.89	n.d	0.22	0.21	0.08	0.18	n.d	0.37	0.02
Glycerolipids			0.57	0.47	0.88	0.83	0.45	0.54	0.68	0.81
1-Monopalmitin, 2TMS	474	40.17	0.57	0.47	0.43	0.83	0.45	0.33	0.68	0.40
2-Palmitoylglycerol, 2TMS	474	43.15	n.d	n.d	n.d	n.d	n.d	0.21	n.d	n.d
Glycerol monostearate, 2TMS	502	43.60	n.d	n.d	0.45	n.d	n.d	n.d	n.d	0.41
Hydroxy acids and derivates			0.76	4.91	4.31	5.99	5.00	7.52	7.25	3.90
Glycolic acid, 2TMS	220	4.10	0.10	0.12	0.08	0.08	0.08	n.d	0.09	0.03
Hydracrylic acid, 2TMS	234	5.36	n.d	n.d	n.d	n.d	n.d	0.13	n.d	n.d
2-Isopropyl-3-ketobutyrate, bis(O-TMS)	288	13.58	n.d	n.d	0.09	n.d	n.d	0.12	n.d	0.14
Lactic Acid, 2TMS	234	3.86	0.24	0.30	0.25	0.14	1.00	2.23	0.18	0.07
Malic acid, 3TMS	350	14.91	0.18	4.30	3.81	5.67	2.44	3.39	4.21	3.60
Glyceric acid, 3TMS	322	10.25	0.13	0.18	0.07	0.11	0.12	0.16	0.11	0.06
Mannonic acid, 4TMS	466	25.81	0.12	n.d	n.d	n.d	1.37	1.48	2.66	n.d
Keto acids and derivates			n.d	n.d	n.d	n.d	n.d	0.46	n.d	n.d
2-Ketobutyric acid, TMS	174	10.94	n.d	n.d	n.d	n.d	n.d	0.13	n.d	n.d
2-Oxovaleric acid	230	9.18	n.d	n.d	n.d	n.d	n.d	0.33	n.d	n.d
Organooxygen compounds			7.82	9.77	12.64	6.90	10.24	16.52	10.66	4.21
1,2,3-Butanetriol, 3TMS	322	6.56	n.d	n.d	0.44	n.d	0.60	1.95	n.d	0.22
2,3-Butanediol, 2TMS	234	3.58	0.38	0.52	n.d	n.d	0.46	0.81	0.28	n.d
Glycerol, 3TMS	308	8.70	7.38	9.25	10.51	6.90	8.99	13.44	8.61	3.08
meso-Erythritol, 4TMS	410	21.16	n.d	n.d	n.d	n.d	n.d	0.21	n.d	0.11
Myo-Inositol, 6TMS	612	30.45	0.06	n.d	0.11	n.d	0.19	0.12	0.17	0.08
Ribitol, 5TMS	512	21.68	n.d	n.d	1.58	n.d	n.d	n.d	1.59	0.72
Quinones			0.34	0.53	0.09	0.16	0.22	0.34	0.25	n.d
Kojic acid, 2TMS	286	19.89	0.34	0.53	0.09	0.16	0.22	0.34	0.25	n.d

* Compound percentages were calculated based on the total number of identified compounds. ** Not detected. RT: retention time. TMS: trimethylsilyl. T + P25: Vortex extraction with precipitation at 25 °C. T + P40: Vortex extraction with precipitation at 40 °C. T + P80: Vortex extraction with precipitation at 80 °C. E + P: Extraction with precipitation. T-P25: Vortex extraction without precipitation at 25 °C. T − P40: Vortex extraction without precipitation at 40 °C. T − P80: Vortex extraction without precipitation at 80 °C. E − P: Extraction without precipitation.

**Table 2 pharmaceuticals-16-01173-t002:** Identification of the constituents of *Plinia cauliflora* extracts using ultra-high-performance liquid chromatography–tandem mass spectrometry (UHPLC-MS/MS) in positive mode.

Compound	Relative Composition (%) *									
	Mw	RT	T + P25	T + P40	T + P80	E + P	T − P25	T − P40	T − P80	E − P *
Flavonoids			35.58	67.73	54.37	18.47	69.04	14.54	11.04	45.15
Quercetin	303	37.18	12.47	21.01	19.84	3.82	31.98	7.33	3.24	16.75
O-hexosyl quercetin	464	30.90	5.36	2.54	2.50	2.23	2.91	1.03	1.42	3.15
O-hexosyl delphinidin	477	28.54	0.40	0.45	0.98	0.14	n.d **	n.d	n.d	n.d
O-hexosyl cyanidin	449	11.57	11.97	30.67	12.43	2.11	27.87	1.16	1.50	11.55
O-deoxyhexosyl quercetin	448	30.38	5.36	8.71	13.15	7.67	6.28	2.30	2.07	9.46
O-deoxyhexosyl myricetin	464	34.24	n.d	4.35	5.47	2.51	n.d	2.73	2.79	4.23
Phenolic acids			0.85	0.56	n.d	0.15	n.d	n.d	n.d	n.d
Syringic acid	198	43.87	0.85	0.56	n.d	0.15	n.d	n.d	n.d	n.d
Tannins and derivates			7.82	15.36	21.78	19.21	5.12	5.36	81.10	4.85
O-galloyl ellagic acid	474	1.54	7.21	10.95	16.19	17.28	n.d	0.30	n.d	n.d
O-hexosyl ellagic acid	480	28.54	0.46	0.45	1.39	0.30	n.d	n.d	n.d	n.d
O-pentosyl ellagic acid	470	32.61	0.15	0.37	0.29	0.24	0.88	0.20	78.87	0.22
Di-O-galloyl hexoside	500	43.50	n.d	n.d	n.d	n.d	n.d	3.39	0.79	1.68
Tri-O-galloyl hexoside	648	32.27	n.d	3.59	3.91	1.39	4.24	1.48	1.44	2.96
Sugars and derivates			58.93	16.36	23.85	62.18	17.25	80.10	7.87	50.00
di-hexoside	342	1.05	57.72	16.01	3.86	17.75	16.03	63.87	7.29	14.56
Hexose	180	1.20	1.20	0.35	19.99	44.42	1.22	16.23	0.58	35.44

* Compound percentages were calculated based on the total number of identified compounds. ** Not detected. RT: retention time. T + P25: Vortex extraction with precipitation at 25 °C. T + P40: Vortex extraction with precipitation at 40 °C. T + P80: Vortex extraction with precipitation at 80 °C. E + P: Extraction with precipitation. T − P25: Vortex extraction without precipitation at 25 °C. T − P40: Vortex extraction without precipitation at 40 °C. T − P80: Vortex extraction without precipitation at 80 °C. E − P: Extraction without precipitation.

**Table 3 pharmaceuticals-16-01173-t003:** Identification of the constituents of *Plinia cauliflora* extracts using ultra-high-performance liquid chromatography–tandem mass spectrometry (UHPLC-MS/MS) in negative mode.

Compound	Relative Composition (%) *									
	Mw	RT	T + P25	T + P40	T + P80	E + P	T − P25	T − P40	T − P80	E − P *
Flavonoids			0.83	1.13	1.22	1.44	0.84	0.63	0.90	1.80
Quercetin	303	12.4	0.13	0.11	0.06	0.04	0.12	0.03	0.03	0.08
O-hexosyl quercetin	464	9.2	0.25	0.22	0.54	0.20	0.03	0.03	0.03	0.14
O-hexosyl cyanidin	449	5.0	0.01	0.32	0.10	0.69	0.06	0.04	0.13	0.89
O-deoxyhexosyl quercetin	448	10.2	0.43	0.49	0.53	0.51	0.63	0.54	0.72	0.69
Phenolic acids			0.25	0.23	0.29	0.49	0.68	0.17	0.23	2.81
Gallic acid	170	2.7	0.25	0.23	0.29	0.49	0.63	0.17	0.23	2.81
Syringic acid	198	9.1	n.d **	n.d	n.d	n.d	0.04	n.d	n.d	n.d
Tannins and derivates			0.16	0.37	0.44	1.18	0.45	0.58	0.33	1.67
Ellagic acid	310	9.4	n.d	n.d	0.01	0.32	0.13	0.22	0.10	0.56
O-cinnamoyl O-galloyl hexoside	470	6.6	0.15	0.15	0.16	0.23	0.15	0.25	0.14	0.25
O-pentosyl ellagic acid	470	9.8	n.d	0.03	0.03	0.03	n.d	n.d	0.01	0.04
O-galloyl ellagic acid	474	12.8	n.d	0.02	0.01	0.11	0.04	0.06	0.02	0.06
HHDP galloyl O-hexoside	802	6.9	n.d	n.d	n.d	0.01	n.d	n.d	n.d	0.01
HHDP di-galloyl O-hexoside	794	7.9	0.00	0.01	0.05	0.34	0.01	0.01	0.01	0.26
di-HHDP O-hexoside	784	7.3	n.d	n.d	n.d	0.02	n.d	n.d	0.01	0.03
Di-O-galloyl hexoside	500	3.3	n.d	n.d	n.d	n.d	0.02	n.d	n.d	0.44
di-HHDP-galloyl O-hexoside	972	4.6	n.d	n.d	n.d	n.d	n.d	0.00	0.01	n.d
HHDP tri-galloyl O-hexoside	966	14.8	n.d	0.04	0.08	0.11	0.10	n.d	n.d	n.d
di-HHDP-galloyl O-hexoside (castalagin/vescalagin isomer)	978	4.6	n.d	0.11	0.10	0.02	0.01	0.03	0.04	0.02
Tri-O-galloyl hexoside	648	3.7	n.d	n.d	n.d	n.d	n.d	n.d	0.01	n.d
Sugars and derivates			18.86	15.28	15.91	21.25	17.57	11.19	1.40	2.08
di-hexoside	342	1.1	1.54	n.d	n.d	n.d	n.d	n.d	n.d	0.22
Hexose	180	1.0	17.32	15.28	15.91	21.25	17.57	11.19	1.40	1.86
Carboxylic acids and derivates			77.35	78.05	76.62	68.60	74.10	79.42	88.80	82.43
Citric acid	192	2.3	77.35	78.05	76.62	68.60	74.10	79.42	88.80	82.43
Alkaloids			2.56	4.92	5.52	7.05	6.36	8.02	8.34	9.22
Quinic acid	203	1.4	2.56	4.92	5.52	7.05	6.36	8.02	8.34	9.22

* Compound percentages were calculated based on the total number of identified compounds. ** Not detected. RT: retention time. T + P25: Vortex extraction with precipitation at 25 °C. T + P40: Vortex extraction with precipitation at 40 °C. T + P80: Vortex extraction with precipitation at 80 °C. E + P: Extraction with precipitation. T − P25: Vortex extraction without precipitation at 25 °C. T − P40: Vortex extraction without precipitation at 40 °C. T − P80: Vortex extraction without precipitation at 80 °C. E − P: Extraction without precipitation.

**Table 4 pharmaceuticals-16-01173-t004:** Quantification of total phenolics (TPC) and total flavonoids (TFC) in *Plinia cauliflora* extracts.

Method	Phenolic Compounds (µg EAG gext^−1^)	Flavonoids (µg QUE gext^−1^)
T + P25	25.54 ± 0.42 f	3.05 ± 0.02 e
T+ P40	45.86 ± 1.07 d	5.39 ± 0.01 c
T + P80	51.72 ± 0.24 c	6.36 ± 0.11 b
E + P	115.59 ± 1.79 a	6.95 ± 0.04 a
T − P25	63.94 ± 0.84 b	4.82 ± 0.08 d
T − P40	38.10 ± 0.36 e	6.34 ± 0.05 b
T − P80	26.92 ± 1.17 f	4.97 ± 0.07 d
E − P	49.08 ± 1.05 c	3.09 ± 0.03 e
Sig.	<0.001	<0.001

Mean ± standard error (n = 3). Means followed by the same letter in a column do not differ significantly according to Duncan’s test (*p* ≤ 0.05). T + P25: Vortex extraction with precipitation at 25 °C. T + P40: Vortex extraction with precipitation at 40 °C. T + P80: Vortex extraction with precipitation at 80 °C. E + P: Extraction with precipitation. T − P25: Vortex extraction without precipitation at 25 °C. T − P40: Vortex extraction without precipitation at 40 °C. T − P80: Vortex extraction without precipitation at 80 °C. E − P: Extraction without precipitation.

**Table 5 pharmaceuticals-16-01173-t005:** Quantification of the scavenging capacity of DPPH and ABTS radicals and the reducing capacity of FRAP complex of *Plinia cauliflora* extracts.

Method	DPPH (μM_Trolox_)	FRAP (µmol_Trolox_ g_ext_^−1^)	ABTS (µmol_Fe_^2+^ g_ext_^−1^)
T + P25	315.88 ± 5.86 d	829.83 ± 9.97 e	1141.00 ± 31.10 cd
T + P40	360.93 ± 6.03 c	1067.38 ± 12.14 b	1092.31 ± 44.27 cd
T + P80	388.66 ± 4.51 b	990.35 ± 4.27 c	1242.42 ± 19.92 b
E + P	489.16 ± 4.51 a	1330.80 ± 5.92 a	1483.12 ± 10.73 a
T − P25	247.72 ± 7.64 e	800.20 ± 8.36 f	1073.38 ± 16.88 d
T − P40	204.98 ± 2.08 f	443.60 ± 8.96 g	843.50 ± 50.08 e
T − P80	334.94 ± 6.80 d	933.25 ± 14.0 d	1173.45 ± 15.59 bc
E − P	333.21 ± 8.96 d	999.51 ± 10.31 c	1484.47 ± 18.19 a
Sig.	<0.001	<0.001	<0.001

Mean ± standard error (n = 3). Means followed by the same letter in a column do not differ significantly according to Duncan’s test (*p* ≤ 0.05). T + P25: Vortex extraction with precipitation at 25 °C. T + P40: Vortex extraction with precipitation at 40 °C. T + P80: Vortex extraction with precipitation at 80 °C. E + P: Extraction with precipitation. T − P25: Vortex extraction without precipitation at 25 °C. T − P40: Vortex extraction without precipitation at 40 °C. T − P80: Vortex extraction without precipitation at 80 °C. E − P: Extraction without precipitation.

**Table 6 pharmaceuticals-16-01173-t006:** Minimum inhibitory concentration (MIC) (mg mL^−1^) of *Plinia cauliflora* extracts.

Method	*B. subitilis*	*E. coli*	*S. aureus*	*P. aeruginosa*	*C. albicans*
T + P25	15.62 ± 0.00	5.20 ± 2.26	20.83 ± 9.02	15.62 ± 0.00	>125
T + P40	20.83 ± 9.02	6.51 ± 2.26	31.25 ± 0.00	15.62 ± 0.00	>125
T + P80	20.83 ± 9.02	7.81 ± 0.00	15.62 ± 0.00	15.62 ± 0.00	>125
E + P	15.62 ± 0.00	7.81 ± 0.00	15.62 ± 0.00	13.02 ± 4.51	>125
T − P25	15.62 ± 0.00	10.41 ± 4.51	20.83 ± 9.02	20.83 ± 9.02	>125
T − P40	26.04 ± 9.02	7.81 ± 0.00	31.25 ± 0.00	10.41 ± 4.51	>125
T − P80	26.04 ± 18.04	52.08 ± 18.04	62.50 ± 0.00	62.50 ± 0.00	>125
E − P	62.50 ± 0.00	62.50 ± 0.00	125.00 ± 0.00	20.83 ± 9.02	>125

Mean ± standard error (n = 3). T + P25: Vortex extraction with precipitation at 25 °C. T + P40: Vortex extraction with precipitation at 40 °C. T + P80: Vortex extraction with precipitation at 80 °C. E + P: Extraction with precipitation. T − P25: Vortex extraction without precipitation at 25 °C. T − P40: Vortex extraction without precipitation at 40 °C. T − P80: Vortex extraction without precipitation at 80 °C. E − P: Extraction without precipitation.

## Data Availability

Data are contained within the article.

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
