# Peer review of "Extraction Methods, Chemical Characterization, and In Vitro Biological Activities of *Plinia cauliflora* (Mart.) Kausel Peels"

_pharmaceuticals, 2023, doi:10.3390/ph16081173_

Round 1

Reviewer 1 Report

The paper is exceptionally well written, with a few minor omissions

1. In Line 39 you wrote liver prevention, I think we need prevention of liver disease or liver damage

2. Row 305 which ethanol, which concentrations?

3. Consider whether you could replace the word infusio with extract in the entire paper

4. 4.3.1. do you have any of the validation parameters?

5. Line 355 you put chapter 1.3. check it out

6. Check the number of decimal places in the result tables. Regardless of the extract, you should have the same number of decimal places for the same compound. Also, check the number of decimal places for SD.

7. Were the chromatographic analyses performed only once? No SD

Reviewer 2 Report

The present study aimed to explore the standard extraction method for P. cauliflora fruit peels to maximize the content of phenolic constituents and evaluate their antioxidant and antimicrobial effects. Although I appreciated authors′ efforts, the present results only displayed preliminary advances and no significant improvements on the bioactive components discovery related fields could be observed. Briefly, this paper is not recommended to accept for publication in Pharmaceuticals. In addition, there are some major comments to be addressed as following.  

1. There were still some typographic, grammar, and format errors to be observed in the text. Authors have to check and revise these errors carefully.

2. The GC/MS and UHPLC-MS/MS analysis were only in the semi-quantitative modes. These data could not reflect the real contents of the compounds in the extracts.

3. The bioactivities were only examined in the extracts level. There were not any single compounds tested for their bioactivity. In addition, there were not any data for positive controls. It is difficult to evaluate the potentials of these extracts.

4. In the References section, the format of some references did not follow the style of this journal. Authors have to check and revise these errors.

There were still some typographic, grammar, and format errors to be observed in the text. Authors have to check and revise these errors carefully.

Reviewer 3 Report

The manuscript in reference describes the chemical and biological study of Plinia cauliflora, oriented to maximize the content of phenolic constituents following a standardized extraction method. Their LC-MS chemical characterization, antioxidant capacity, and antimicrobial effects were also evaluated. Although the manuscript has relevant elements and passages, several points need to be addressed before further consideration:

1.     The abstract needs to be improved since its passages are challenging to be followed due to an unorganized sequence and mixing of several ideas. A detailed organization is recommended.

2.     Lines 32-42: These paragraphs can be merged for better presentation.

3.     Lines 48-58: This paragraph must be improved since the information is mixed and challenging to follow.

4.     Lines 59-61: Revise the objective order since the main aim is to maximize the content of phenolic constituents by evaluating various extraction methods to chemically and biologically characterize the resulting extracts and observe related variations.

5.     Lines 67-69 and Figure: why were these extraction procedures implemented? Logistics? Any ruled extraction procedures? The explanations and justification for using such extraction protocols must be appropriately provided for readers. For instance, why were these temperatures (40 and 80°C) selected? Any previous standardization?

6.     Line 72 and 301: Time for vortexing?

7.     Figure 1: Inferential statistics need to be applied to these data, as performed for the data in Table 4. How many technical or biological replicates were included to get these data? This information is required to be provided to ensure data reproducibility. In this regard, data dispersion in terms of SD or SEM must be added to improve the data quality and discuss the significant differences between extraction protocols.

8.     Line 78: The metabolite identification by MS from GC-MS and LC-MS recordings must be clearly explained to avoid confusing interpretations by readers. There is no explanation of how the authors find the structures. Therefore, a paragraph explaining metabolite identification and the level (i.e., 1,2,3, etc.) must be provided from the data of both analytical platforms.

9.     Table 3: The way of obtaining the relative abundance must be provided. In addition, the accurate mass and error of each metabolite in this table must be informed for readers.

10.  Table 6: How many replicates?

11.  Discussion: This section needs to be revised since some passages involve a compilation of results. Detailed scrutiny is recommended.

12.  Line 309: This reference is not in the correct format citation. Be consistent throughout the manuscript.

13.  Line 322: The type of analyzer in the MSD must be informed.

14.  Line 330: Linear retention indexes must be added to ensure correct metabolite identification since the only EIMS comparison with libraries is inadequate for proper identification. This data must also be added in the corresponding sections of the manuscript.

15.  Line 346: This information about MS is repeated (line 333).

16.  Line 367: The antioxidant capacity assays are not related to in vitro trials. They just constitute an in-tube/in-well chemical assay. In addition, they are related to antioxidant capacity (thermodynamically tested) instead of antioxidant activity (kinetically tested). Be consistent throughout the manuscript.

17.  Line 394: Verification of data normality is missing to justify using parametric tests such as ANOVA and DUNCAN.

18.  The conclusion section needs to be also improved since it is very general and laconically organized, and no conceptual findings from the mechanistic point of view are not provided.

Detailed scrutiny should be performed throughout the manuscript to look for grammar, stylistic, and even typos issues. A language editing service is therefore recommended.

Reviewer 4 Report

The MS entitled " Extraction methods, chemical characterization, and in vitro biological activities of Plinia cauliflora (Mart.) Kausel peels" was thoroughly reviewed. The MS is well composed, supported by relevant literature and the results are systematically presented. The authors have discussed about the different extraction methods and the chemical composition of Kausel extracts as well as their antioxidant and antimicrobial activities. The use of GCMS and HPLC MS is well defined.

some of my queries are as under.

1. The scientific names of plants/organisms should be in italic throughout the MS. The literature should be updated.

2. In Total phenolic content assay, usually 517 nm is applied as wave length maxima. kindly mention the wavelength.

4. The expression should be in mg EAG gext-1 rather than µg EAG gext-1) (standard).

5. Provide FID/total ion chromatogram for table 1.

6. Line 296-310. How the precipitate with ethanol was formed? And how was the ppt obtained? Include the process in the said lines.

7.  in GCMS analysis, what kind of derivatization agent was used? Since the extract usually contains non-volatile compounds.

8. why need of GCMS when HPLC-MS was used for detail analysis? Were there any individual standard run along with the sample?

 Extensive editing of English language required.

Round 2

Reviewer 2 Report

I had gone through the revised manuscript and my comments were attached herein. Although authors had provided some responses as compared with the previous version, I still felt that the present manuscript did not meet the minimum criteria of Pharmaceuticals. The characterization of significant constituents was limited and the bioactivities were not examined in the molecular level. There was not any significant improvement of the new drug discovery. Conclusively, I stood on my previous point that this manuscript could not be accepted for publication.

Reviewer 3 Report

The authors addressed my comments adequately so that the manuscript can be considered further.

Reviewer 4 Report

The suggestions have been incorporated and the quality of manuscript is improved.